## RESEARCH CULTURE

# Career choices of underrepresented and female postdocs in the biomedical sciences

**Abstract** The lack of diversity among faculty at universities and medical schools in the United States is a matter of growing concern. However, the factors that influence the career choices of underrepresented minority and female postdoctoral researchers have received relatively little attention. Here we report the results of a survey of 1284 postdocs working in the biomedical sciences in the US. Our findings highlight possible reasons why some underrepresented minority and female postdocs choose not to pursue careers in academic research, and suggest interventions that could be taken in the early stages of postdoctoral training to prevent this attrition of underrepresented groups.

**W MARCUS LAMBERT\*, MARTIN T WELLS, MATTHEW F CIPRIANO, JACOB N SNEVA, JUANITA A MORRIS AND LINNIE M GOLIGHTLY**

**\*For correspondence:** wil2009@
med.cornell.edu

**Competing interests:** The authors declare that no competing interests exist.

## Introduction

Despite moderate success in increasing the diversity of doctoral students in the biomedical sciences, the proportion of underrepresented minority (URM) faculty in the United States has not increased (*Gibbs et al., 2016*; *Gumpertz et al., 2017*; *Meyers et al., 2018*). At universities in the US, URM researchers make up a mere 3.5% of the faculty in the life sciences and just 6.3% of the faculty in basic science departments at medical schools. Women also remain underrepresented, accounting for only 39% of the faculty in the life sciences at universities and 35% of the faculty in basic science departments at medical schools (*Association of American Medical Colleges, 2017*; *National Science Foundation, 2019*). The continued underrepresentation of female and URM faculty in the US will have far-reaching implications on the recruitment, mentorship, and retention of trainees (*Xierali et al., 2016*), as well as the quality of science produced by academic institutions (*Antonio et al., 2004*; *Campbell et al., 2013*).

Fixing this disparity requires a multifaceted approach. Simply 'feeding the pipeline' by increasing the number doctoral students from underrepresented groups (women and racial/ethnic minorities) is insufficient to increase faculty diversity. It has been shown that the number of scientists from URM backgrounds who are hired as assistant professors in US medical schools is not significantly correlated with the number of potential candidates (i.e., the number of URM PhD graduates; *Gibbs et al., 2016*). In fact, the proportion of the URM candidate pool hired into faculty positions decreases each year. Addressing the dearth of faculty diversity will involve an assessment of faculty recruitment processes, institutional hiring practices and overall postdoctoral interest in academic positions, which has declined among underrepresented groups.

There is evidence that the number of PhD students interested in pursuing a career in academic research declines as they progress in their graduate training, and it has been shown that more female and URM trainees are opting for careers

outside of academia (*Fuhrmann et al., 2011*; *Gibbs et al., 2014*; *Gibbs et al., 2015*; *Layton et al., 2016*; *Roach and Sauermann, 2017*; *Sauermann and Roach, 2012*). It is estimated that only 21% of postdocs move into tenure-track faculty positions in the biomedical sciences, though this percentage increases to an estimated 37% when other full-time faculty or equivalent positions are included (*Silva et al., 2016*; *Table 1*).

PhDs are entering postdoctoral training with interests in a broad array of career paths including academia, industry or for-profit organizations, and non-research positions (*Mathur et al., 2018*; *Sauermann and Roach, 2016*). From their postdoctoral training they develop transferrable skills that are crucial to success in a wide range of careers (*Sinche et al., 2017*). This suggests that nonacademic careers are not simply 'alternative' career options. Yet, with 70% of biomedical PhDs initiating postdoctoral training after they graduate, it is unclear how many have developed a clear career goal and at what stage (*Gibbs and Griffin, 2013*).

Gibbs et al. have noted that high interest in academia at the beginning of the PhD, high research self-efficacy, and higher first-author publication rates were positively associated with interest in faculty careers at research-universities (*Gibbs et al., 2014*). Further work by St. Clair et al. examined the strategies and resources students use to prepare for a broad job market: this revealed that students seeking nonacademic careers have lower confidence in the career search process and seek less career advice from their advisors (*St. Clair et al., 2017*). Roach and Sauermann also note that the declining interest in academic careers is not driven by concern about job availability or funding. Instead, they found it is, in part, due to students' changing preferences

for specific job attributes not found in an academic research career (*Roach and Sauermann, 2017*). For example, students who lose interest in academia later in their PhD training were found to have stronger interests in job attributes such as commercialization rather than basic research.

While these reports address the career choices and preferences of PhD students, the factors that influence the career goals of postdocs are less well understood. Moreover, those who have started to explore this topic acknowledge the differences between graduate students and postdocs (*Gibbs et al., 2015*; *St. Clair et al., 2017*). A few studies have begun to explore the factors that shape career interests among PhD-trained scientists, and have shown that both gender and underrepresentation can play a distinct role in which factors are important for shaping career interests and retention (*Gibbs et al., 2015*; *Hechtman et al., 2018*; *Layton et al., 2016*; *Martinez et al., 2018*; *McConnell et al., 2018*).

In this article, we report the results of a survey that explores whether biomedical science postdocs in the US want to pursue faculty positions given the changing career landscape, and how to increase diversity in academia if underrepresented PhD students are losing interest in academic positions. We hypothesize that there are a significant number of postdocs with aspirations of becoming faculty members in academia, but many seek nonacademic careers due to uncharacterized influences. Using several behavioral science theories as a foundation for the exploration of career choice, we have characterized: i) the interests of postdocs in academic research careers by gender and underrepresentation; ii) research outcome expectations and research self-efficacy among female and URM postdocs; iii) the top predictors for pursuing a

**Table 1.** Percentage of PhDs and postdocs who enter academic positions.

| Population | Outcome | Year(s) | Citation |
|---|---|---|---|
| Biomedical PhDs who enter academia (research/teaching) | 43% | 2012 | (*Biomedical Research Workforce Working Group, 2012*) |
| Biomedical PhDs who enter tenure-track | 23% | 2012 | (*Biomedical Research Workforce Working Group, 2012*) |
| Life Science PhDs who enter tenure-track within 5 years | 8.1% | 2015 | (*National Science Foundation, 2018*) |
| Biomedical Postdocs who enter tenure-track | 27.4% | 1980–2003 | (*Kahn and Ginther, 2017*) |
| Biomedical Postdocs who enter tenure-track | 21% | 2013 | (*Kahn and Ginther, 2017*) |
| UCSF Postdocs who enter faculty positions | 37% | 2000–2013 | (*Silva et al., 2016*) |

research career by gender and underrepresentation.

## Results

### Academia remains a top choice for biomedical postdocs

To better understand the intended career paths of postdoctoral fellows, survey participants were asked to rank the following careers on a five-point Likert-type scale from *Most likely to pursue* to *Least likely to pursue*: (1) Academic (Faculty), Research-Intensive; (2) Academic (Faculty), Teaching-Intensive; (3) Other Research-Intensive (e.g. Industry); (4) Non-Research, Science-Related; and (5) Non-Science Related. Despite the well-documented odds of obtaining an academic faculty position, 59% of biomedical and biological postdocs who were sampled identified faculty positions in academia as their top career choice (48% research-intensive and 11% teaching-intensive; *Figure 1a*). Furthermore, three-quarters of postdocs intend to pursue research careers, whether in academia or other research sectors such as industry.

Career intentions are also reported by gender and underrepresentation (*Figure 1b*). While we observed no difference between URM and well-represented female postdocs seeking a research-intensive faculty position, fewer URM males intend to pursue research faculty positions. In fact, the odds of intending to pursue a non-research or non-science career are twice as high for URM male postdocs (n = 55) compared to other postdocs (OR 1.97, 95% CI 1.035, 3.75, p<0.05), a statistic in line with previous analyses on African American males in science and medicine (*National Academies of Sciences, Engineering, & Medicine, 2018*). The other notable difference is interest in teaching, with more female postdocs and URM male postdocs interested in pursuing teaching-intensive faculty careers (*Figure 1b*). For simplification, reference to *academic* careers in the remaining figures will focus on research-intensive faculty positions and exclude teaching-intensive positions, since teaching only comprises 11% of this pool. Due to the small sample of male URM respondents, unless stated otherwise, from here on the data is grouped either based on gender or minority status: this means that female respondents includes both well-represented (WR) and URM postdocs, and URM respondents includes both male and female postdocs.

To better understand the pattern of interest in academic careers as trainees progress in their postdoc, we determined the percentage of respondents intending to pursue academia across 7 years (*Figure 1c*). The percentage of postdocs intending to pursue academia decline between years 1 and 3, and rises 11% by year 7. Despite a steady decline in the number of respondents each year, intention to pursue academia increased between years 3 and 7. This increase in the percentage of postdocs pursuing academia is likely due to the exiting of postdocs pursuing nonacademic careers, as seen in other studies (*Silva et al., 2016*). Female and URM postdocs also show a decline in the first two years: however, well-represented male postdocs increase during this period, making a decline in year 4 and the most significant decline by year 6 (when 5 year term limits take effect for some institutions) (*Figure 1d*; *Figure 1—figure supplement 1a,b*).

We then asked respondents who intend to pursue nonacademic careers (including teaching-intensive careers) how their commitment to academia has changed since starting their postdoctoral position. 54% of early postdocs (<3 years) were less committed to academia since starting their postdoc, while 37% seemed to enter postdoctoral training already more committed to a nonacademic career (*Figure 1—figure supplement 1c*). We also found that a larger percentage (65%) of late-stage postdocs (>3 years) were shown to be less committed to academia. This correlates with other reports demonstrating that the likelihood of postdocs obtaining an academic career diminish with longer training periods (*van der Weijden et al., 2016*). Taken together, these data suggest that: i) career intentions do not remain static; ii) the first two years of training are the most critical for retaining academic career interests for female and URM postdocs.

We also determined the percentage of respondents from highly-ranked life science research institutions in the US based on counts of high-quality research outputs between January 1, 2017 and December 31, 2017 according to *Nature Index* (*Supplementary file 2*). 47% of respondents who intend to leave academia and 57% of respondents pursuing academic careers are conducting their postdoc at one of the top 25 life science research institutions (*Figure 1—figure supplement 1d*). 72% of nonacademic-bound postdocs and 78% of academic-bound postdocs are based at one of the top 50-ranked institutions. Thus, the majority of respondents are from highly-ranked US institutions, perhaps

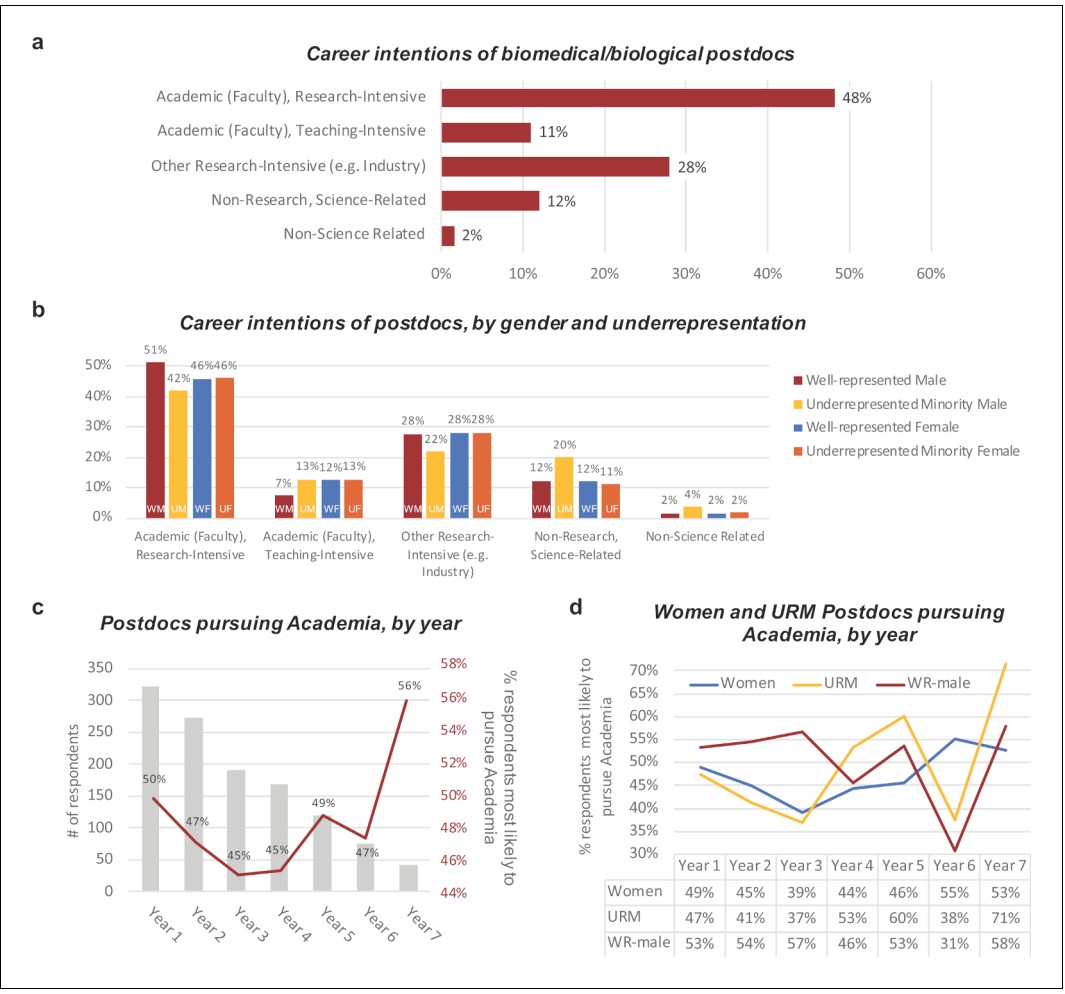

**Figure 1.** Career intentions of biomedical postdocs. (a) Percentage of respondents likely to pursue five different career paths. (b) Percentage of respondents likely to pursue these career paths broken down by gender and well-represented/underrepresented minority. Underrepresented minorities include the racial categories of American Indian or Alaska Native, Black or African American, Native Hawaiian or other Pacific Islander and/or the ethnicity of Hispanic or Latino. Well-represented respondents identified as Asian or White and Non-Hispanic or Latino. (c) Percentage of postdocs in years 1 to 7 of their training who are likely to pursue a research position in academia (red line, right vertical axis). The number of survey respondents in each year is represented by the grey bars (left vertical axis). (d) Percentage of women (blue), underrepresented minority (yellow; URM), and well-represented male (red; WR-male) postdocs most likely to pursue a research career in academia in years 1 to 7 of their training (the number of respondents in each category are shown in *Figure 1—figure supplement 1a*).

The online version of this article includes the following source data and figure supplement(s) for figure 1:

**Source data 1.** Source data on career intentions of postdocs.

**Figure supplement 1.** Characteristics of postdoc respondents.

**Figure supplement 1—source data 1.** Source data on change in commitment to academia for nonacademic-bound postdocs.

skewing the number of those who want to pursue a career in academic research. However, the differences in institutional ranking do not fully account for the differences in career intention.

## Many of the most productive postdocs opt out of academia

As a metric of scientific productivity, we asked participants to report their total number of publications, their number of first-author publications, and the highest impact factor of any one of their publications. Postdocs who intend to pursue academic research careers (irrespective of gender or

underrepresentation) produced significantly more publications (9 vs. 7, p<0.001), more first-author publications (4 vs. 3, p<0.001), and had a higher first-author publication rate (0.56 vs. 0.42, p<0.001), calculated by dividing the number of first author publications produced by the number of years since they started their PhD (*Table 2*). This rather crude measure of productivity was easy to self-report and has been shown to be predictive of becoming a principal investigator (PI) in academia (*van Dijk et al., 2014*).

The highest impact factor of the journals in which participants have published was not significantly different between those choosing academic vs. nonacademic careers. Interestingly, there were 144 respondents (11%) whose total publications and highest journal impact factor were in the 75th percentile or above. 60% of those respondents intend to pursue academic research faculty positions, while 40% are choosing different career tracks, more commonly research-intensive positions outside of academia (*Figure 1—figure supplement 1e*). Only a very small percentage are intending to pursue a non-science career track or a teaching-intensive faculty position. These results suggest that a significant portion of even the most productive postdocs, regardless of gender or underrepresentation, are opting out of pursuing an academic career.

### Understanding what influences career choice

To understand the factors that contribute to the decision whether or not to pursue an academic research career, we posed a series of statements to participants (*Supplementary file 1*). From these statements we identified the most influential factors for those who intend to pursue a research position in academia vs. those who are most likely to choose teaching or nonacademic career paths (this includes teaching-intensive faculty positions, nonacademic research positions such as industry, non-research but science-related positions, and non-science related positions). As expected, 'job prospects' was the most cited reason for those who are opting for careers in sectors other than academia (*Figure 2a*). The differences between early and late postdocs regarding 'the factors that influence career intentions' were not large enough to alter the trends shown throughout the manuscript.

Participants also identified 'financial security' and 'responsibility to family' (significant other/ spouse, children, and/or other dependents) as top factors. 56% of postdoc respondents intending to pursue academia are married. Significantly more postdocs pursuing nonacademic careers cited 'responsibility to family' as an influential factor in their career choice (57% vs. 37%). Similarly, 28% and 25% of academic-bound and non-academic-bound respondents, respectively, indicated having children or other dependents (not a significant difference, p=0.57).

### Postdocs lack career guidance from mentors

In order to better understand the role of mentorship in career intentions, we asked participants to rate how guidance from their group leader or principal investigator (PI) influences their career choice. 58% of those intending to pursue an academic career agreed that career

**Table 2.** Median number of publications, publication rates, and journal impact factors by career-intention, gender, and underrepresentation.

Survey participants were asked to report the total number of publications in which they are listed as an author, the total number of first-author publications, and the highest impact factor of the journals in which they have published. First-author publication rate was calculated by dividing the number of first-author publications by the number of years since they started their PhD. Median values with the interquartile range are shown. **p<0.001, *p<0.01, #p<0.05.

| | Total no. of publications N = 1282 | No. of first-author publications N = 1282 | First-author publication rate N = 1270 | Highest impact factor N = 1275 | Total |
|---|---|---|---|---|---|
| All | 8.0 [5.0;12.0] | 3.0 [2.0;5.0] | 0.50 [0.27;0.82] | 8.50 [4.94;13.2] | 1284 |
| Academic-bound | 9.0 [5.0;13.0] | 4.0 [2.0;6.0] | 0.56 [0.33;1.01] | 8.90 [4.90;14.0] | 595 |
| Nonacademic-bound | 7.0 [4.0;10.0]** | 3.0 [2.0;4.0]** | 0.42 [0.23;0.68]** | 8.29 [4.95;12.1] | 657 |
| Male | 8.0 [5.0;13.0] | 3.0 [2.0;6.0] | 0.51 [0.29;0.92] | 9.10 [5.00;13.9] | 485 |
| Female | 7.0 [5.0;11.0]** | 3.0 [2.0;5.0] | 0.47 [0.26;0.77] | 8.20 [4.72;12.8]# | 774 |
| Well-represented | 8.0 [5.0;12.0] | 3.0 [2.0;5.0] | 0.51 [0.29;0.85] | 8.82 [4.97;13.5] | 1110 |
| Underrepresented Minority | 7.0 [4.0;10.0]* | 2.0 [2.0;4.0]** | 0.38 [0.22;0.70]* | 6.57 [4.80;12.0]# | 174 |

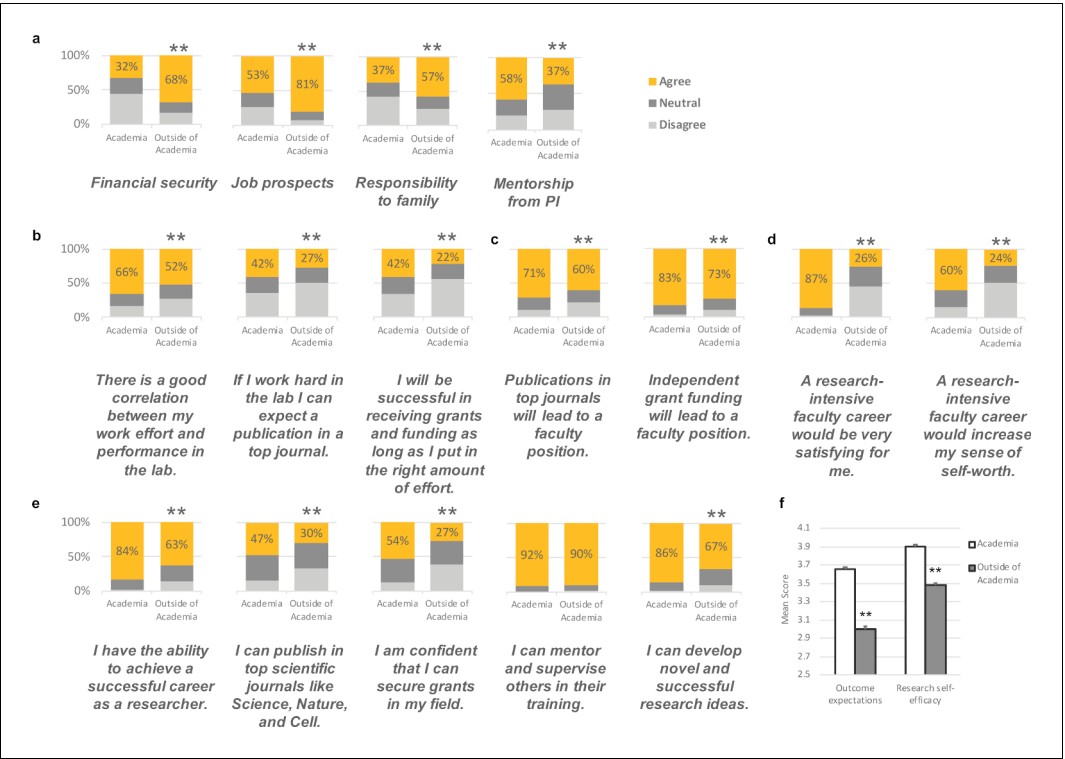

**Figure 2.** Factors that influence the career choices of postdocs. Respondents were asked to rate factors that influenced their career choice on a five-point Likert-type scale (strongly disagree, disagree, neutral, agree, and strongly agree). For the purpose of the figures, *agree* represents both *strongly agree* and *agree* responses, and *disagree* represents both *strongly disagree* and *disagree* responses. **p<0.001. (a) Factors that showed the greatest difference between postdocs intending to pursue academic research and those who do not. The percent of academic-bound and nonacademic-bound postdocs responding to statements relating to (b) expectancy (will my effort lead to high performance?); (c) instrumentality (will performance lead to desirable outcomes?); (d) valence (do I find the outcomes desirable?); and (e) research self-efficacy. The answers to seven of these questions were then used to calculate scores for outcome expectations (f, left), and the answers to the other five questions were used to calculate scores for research self-efficacy (f, right): in both cases the mean score for academic-bound postdocs (white column) was significantly higher than the mean score for nonacademic-bound postdocs (grey column). Cronbach's alpha was used to assess internal consistency: outcome expectations scale has an alpha of 0.73 (with a 95% CI of 0.71—0.75, p<0.0001), and the research self-efficacy scale has an alpha of 0.79 (with a 95% CI of 0.77—0.81, p<0.0001). See Methods for more information on scale development, construct and content validity.

guidance from their PI was influential in their career decision (*Figure 2a*). The percentage was significantly lower at 37% for those pursuing nonacademic positions. We also asked about mentorship from those other than their PI. About half of the postdocs surveyed indicated they lacked career guidance from other mentors, with no significant difference between academic and nonacademic-bound respondents.

### Lower expectations about research outcomes correlate with nonacademic career choice

As implied by the social cognitive career theory, career interests are maintained by positive self-

efficacy and outcome expectations (*Lent et al., 1994*; *Lent et al., 2017*). *Self-efficacy* refers to the belief in one's own ability to succeed at a particular behavior, and *outcome expectations* refer to the consequences or outcomes of performing a particular behavior. Thus, we designed two scales focusing on research outcome expectations and research self-efficacy (*Supplementary file 1*). The outcome expectations scale was based on Vroom's expectancy theory for motivation (*Vroom, 1964*), which posits that (1) *expectancy* (will my effort lead to high performance?), (2) *instrumentality* (will performance lead to desirable outcomes?), and (3) *valence* (do I find the outcomes desirable?) lead

to a motivational force. If any one of these parts are low, then it decreases the motivational force for making behavioral choices. We formulated questions on expectancy (*Figure 2b*), instrumentality (*Figure 2c*), and valence (*Figure 2d*), and found that postdocs who are pursuing academic careers score significantly higher on our outcome expectations scale than those pursuing nonacademic career tracks (3.66 vs. 3.00, p<0.001; Cronbach's Alpha = 0.73 (95% CI = 0.71, 0.75), p<0.0001). This suggests that significantly more postdocs pursuing nonacademic paths: (1) don't expect that their work and effort in the laboratory will generate top publications and grants, (2) don't expect that their publications and grants will lead to faculty positions, and (3) don't value an academic research career in the same way. Thus, these postdocs are presumed to have a lower motivation to achieve academic research careers.

### Nonacademic-bound postdocs consider themselves less able to perform tasks associated with succeeding in academia

Since social cognitive career theory also outlines the importance of self-efficacy to career goals, we designed a self-efficacy scale specifically for research independence in the biomedical sciences. Research self-efficacy refers to an individual's personal beliefs about their ability to perform particular behaviors or courses of action that will achieve a research career, such as publishing in top journals, securing grants, mentoring others, and developing novel research ideas. 63% of postdocs choosing careers outside of research academia felt as if they had the ability to achieve a successful career as a researcher compared to 84% of postdocs seeking research positions in academia (*Figure 2e*). Of the 63%, approximately half are seeking research careers in sectors such as industry, suggesting that the lower self-efficacy is not unique to one career path. Self-efficacy for publications and grants were equally poor with only 30% of respondents who are pursuing nonacademic careers confident in their ability to publish in top journals or secure grants in their field. Overall, postdocs who sought academic careers had higher research self-efficacy than postdocs pursuing other paths (3.9 vs. 3.48, p<0.001; Cronbach's Alpha = 0.79 (95% CI = 0.77, 0.81), p<0.0001; *Figure 2f*).

### Outcome expectations and self-efficacy among female and URM postdocs

To better understand the differences in research outcome expectations and research self-efficacy across demographics we determined overall outcome expectations and self-efficacy scores for female and male survey respondents. Male postdocs scored higher on both the research outcome expectations and research self-efficacy scales with scores of 3.42 and 3.8 (p<0.001), respectively (*Figure 3a*). Female postdoc scores were significantly lower at 3.24 and 3.61, respectively. While the percentage of female and male respondents did not differ for the statement, "I have the ability to achieve a successful career as a researcher", a lower percentage of female postdocs agreed that they could publish in top journals (33% vs. 47%) or secure grants in their field (36% vs. 46%).

Interestingly, there was no significant difference in outcome expectations between URM and well-represented postdocs, however, URM postdocs exhibited a slightly higher research self-efficacy score (*Figure 3b*). While not significant after adjusting for multiple comparisons, this result does point to an interesting consideration that URM postdocs, particularly URM males (*Figure 3—figure supplement 1*), have high research self-efficacy despite being more likely to pursue non-research careers. These findings are similar to work by Gibbs et al. which showed a higher percentage of URM males were confident in their research abilities (*Gibbs et al., 2015*).

We then considered the number of total and first-author publications (*Figure 3c*) and the first-author publication rates (*Figure 3d*) as a marker for productivity. URM postdocs and female postdocs reported lower total numbers of publications. URM postdocs also reported lower numbers of first-author publications and lower first-author publication rates compared to well-represented postdocs.

From *Table 3*, the negative binomial regression suggests a significant positive relationship between research self-efficacy and the number of first-author publications, with a 15% increase in the number of first-author publications for every addition point on the self-efficacy scale. A linear regression analysis (fit on the log scale) suggests a significant positive relationship between research self-efficacy and first-author publication rates, with a 21% increase in the first-author publication rate for every additional point on the self-efficacy scale.

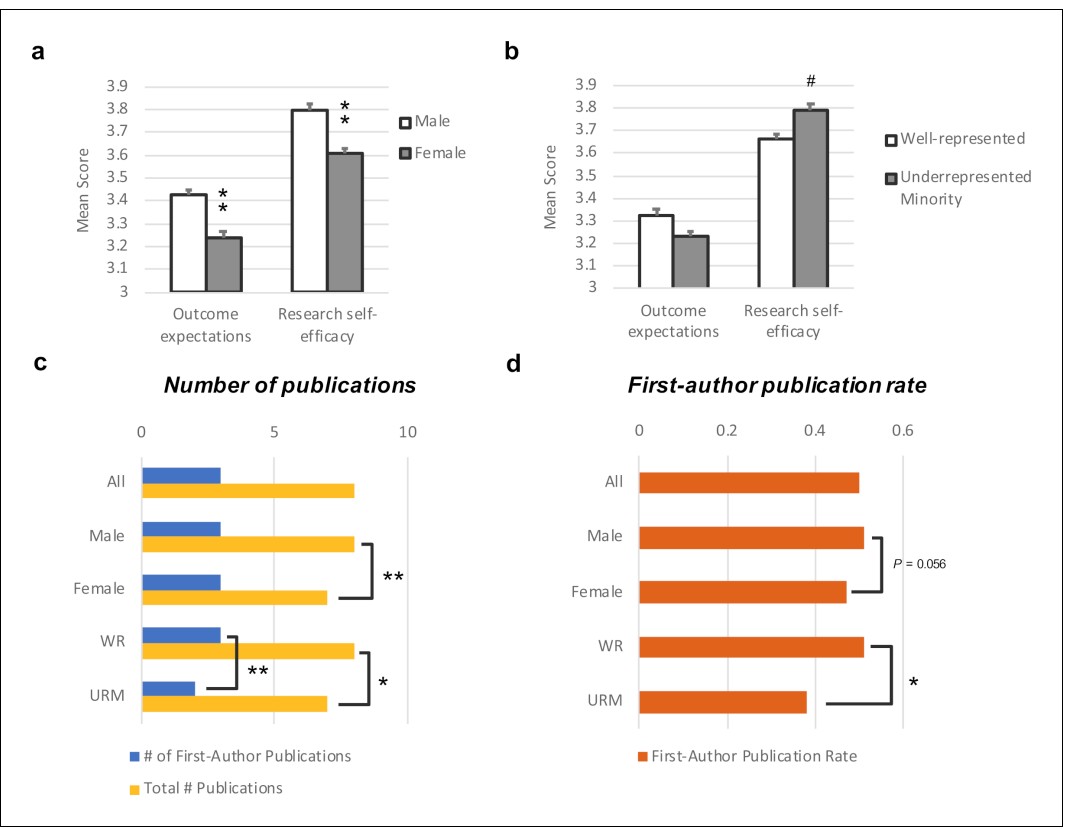

**Figure 3.** Self-efficacy and publication scores differ depending on gender and representation. Results of the outcome expectations and self-efficacy scales are compared by (**a**) gender and (**b**) well-represented/ underrepresented minority respondents. (**c**) The total number of publications (yellow) and first-author publications (blue). (**d**) The first-author publication rate (based on the number of years since starting a PhD) for male, female, URM and WR postdocs. URM and WR groups includes both female and male postdocs. **p<0.001, *p<0.01, #p<0.05.

The online version of this article includes the following source data and figure supplement(s) for figure 3:

**Figure supplement 1.** Differences in outcome expectations and research self-efficacy by gender and underrepresentation.

**Figure supplement 1—source data 1.** Source data on outcome expectations and research self-efficacy.

We also assessed the relationship between both the number of first-author publications and first-author publication rates and research self-efficacy for male, female and URM postdoc only subgroups. Although female postdocs reported lower research self-efficacy than male postdocs, the expected number of first-author publications and the first-author publication rate both increase for women as research self-efficacy increases (p<0.0001). Similarly, for URM postdocs, the expected first-author publication rate increases for URM postdocs with research self-efficacy (p=0.048). There is no significant interaction effect from gender and URM (p=0.371) in *Table 3* or in our other regression models. It is likely that there are too few URM postdocs to see a significant effect, a phenomenon observed

by other researchers recruiting URM postdocs in the biomedical sciences (*Gibbs et al., 2015*).

## Factors that predict the career choice of postdocs

In order to discover the best predictors of career intention, we conducted ordered logit regression analysis of the factors thought to play an influential role in career choice based upon participant responses. A positive sense of 'self-worth' (how you value yourself) and 'career mentorship' from the PI were the top positive predictors for postdocs intending to pursue a research-intensive faculty position (*Figure 4a*). Interestingly, 'financial security' was the top negative predictor, suggesting the less that postdocs see financial security as an influential factor

**Table 3.** Research self-efficacy predicts publications for women and URM postdocs.

Incidence rate ratios (IRR) of first-author publications and relative mean change of first-author publication rates were calculated for respondents who identified as female or underrepresented minority (n = 1,211). The incident rate ratio (IRR) represents the change in the dependent variable in terms of a percentage increase or decrease, with the precise percentage determined by the amount the IRR is either above or below 1. In the pooled sample the IRR for female vs. male (URM vs. WR) is 0.87 (0.82), so there is a 13% (18%) decrease in the first author publications for females vs, males (URM vs. WR). Conversely, there was a 15% increase in the first author publications for every addition point on the self-efficacy scale. The relative mean change gives the percent increase (or decrease) in the response for every one-unit increase in the independent variable. In the pooled sample the percent decrease of first author publication rate for female vs. male (URM vs. WR) is 5% (20%). Conversely, there was a 21% increase in the first author publication rate for every addition point on the self-efficacy scale. Incidence rate ratios or relative mean change of research self-efficacy by subgroups of those identifying as male (n = 463), female (n = 748), well-represented (WR; n = 1,043) and underrepresented minority (URM; n = 168) were also calculated (last four rows of the table). All four subgroups showed a similar increase in the two dependent variables with respect to self-efficacy.

| | First-author publications | | | First-author publication rate | | |
|---|---|---|---|---|---|---|
| | IRR | 95% CI | p-value | Mean Change | 95% CI | p-value |
| Pooled Sample | | | | | | |
| Female vs. Male | **0.87** | 0.81, 0.94 | 0.001 | **0.95** | 0.84, 1.07 | 0.27 |
| URM vs. WR | **0.82** | 0.73, 0.91 | 0.001 | **0.80** | 0.67, 0.95 | 0.012 |
| Research self-efficacy | **1.15** | 1.09, 1.21 | 0.000 | **1.21** | 1.13, 1.31 | 0.000 |
| Subgroups | | | | | | |
| Research self-efficacy for *men only* | **1.20** | 1.09, 1.31 | 0.000 | **1.23** | 1.08, 1.42 | 0.002 |
| Research self-efficacy for *women only* | **1.12** | 1.05, 1.19 | 0.001 | **1.21** | 1.09, 1.32 | 0.000 |
| Research self-efficacy for *WR only* | **1.15** | 1.09, 1.23 | 0.000 | **1.23** | 1.17, 1.31 | 0.000 |
| Research self-efficacy for *URM only* | **1.11** | 0.95, 1.28 | 0.196 | **1.22** | 1, 1.49 | 0.048 |

in their career choice, the more likely they will choose to pursue a research-intensive academic career. Similarly, 'career mentorship' and 'financial security' were top predictors for teaching-intensive positions (*Figure 4b*). As one might expect, other research-intensive positions (i.e., industry), were best predicted by a positive regard for 'financial security' and low 'confidence in securing grants' (*Figure 4c*). A low sense of 'self-worth' and 'career mentorship' were weakly predictive of choosing a research-intensive career path outside of academia, such as industry (*Figure 4c*) and strongly predictive of choosing a non-research career (*Figure 4d*).

The top predictors of a research-intensive faculty position for URM postdocs (*Figure 5a*) were tied to positive self-worth and high self-efficacy around securing grants. The most notable differences among well-represented and URM postdocs were: i) their expectations about associating with people who they value in their future career; and ii) career mentorship (specifically, career guidance from their PI; *Figure 5a, b*). More well-represented postdocs choosing careers in academia felt that they would be able to associate with people that they valued most

in academia. This, in fact, predicted academic career intentions for well-represented postdocs. Conversely, significantly fewer URM postdocs felt that they would be able to associate with people that they value, and this was associated with the trend toward choosing nonacademic careers. Similarly, for career mentorship, well-represented postdocs who receive influential career guidance from their PI are more likely to choose academic research careers. In contrast, this was not true for URM postdocs for whom career mentorship from their PI was not a predictor for academic career intentions.

The best predictor for women intending to pursue an academic research position was also positive self-worth. Female postdocs, however, exhibited higher predictive values for self-efficacy (particularly in novel ideas and grants) and career mentorship than male postdocs pursuing research-intensive faculty positions (*Figure 6a*). Low self-worth and low career self-efficacy were most predictive of non-research careers (*Figure 6b*). Lifestyle was a negative predictor, suggesting differing perspectives or realities of work-life balance. Similar to URM postdocs, if female postdocs agreed that pursuing a

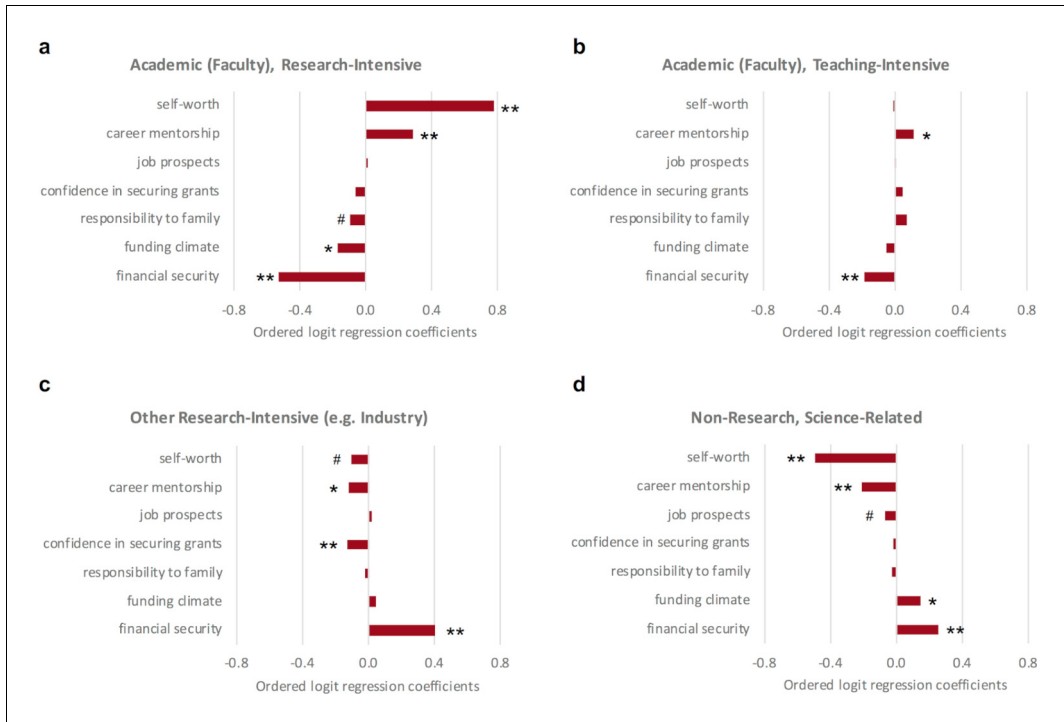

**Figure 4.** Factors that predict a postdoc's choice of career. Ordered logit regression analysis was applied to respondents' answers to see which factors can predict the career intention of postdocs pursuing (**a**) research-intensive, (**b**) academic (faculty), teaching-intensive, (**c**) other research-intensive (e.g. industry), and (**d**) non-research, science-related positions. The higher level of self-worth that postdocs have and the more mentorship they receive from their PI, the more likely they are to pursue a research-intensive career in academia; the more that financial security is an influential factor to their career choice, the more likely postdocs will pursue 'other research intensive (e.g., industry)' and 'non-research; science-related' career paths. The results of the analysis are discussed in more detail in the main text. p-values represent whether the factor is significant with respect to the intention (ranging from least likely to most likely) to pursue the career path listed in the figure sub-header. **p<0.001, *p<0.01, #p<0.05.

research-intensive faculty position would enable them to associate with the kind of people that they value most, it did not significantly change the likelihood that they would pursue a research-intensive faculty position. Whereas for men, the more that they expected to associate with the kind of people that they valued most, the more likely they would pursue a research faculty position. These results suggest a strong consideration for the role of self-worth, self-efficacy, community, lifestyle, and job prospects as predictors for deciding to pursue research-related careers. Moreover, the data suggests that financial concerns could be discouraging postdoctoral fellows from pursuing research-intensive faculty positions.

### URM postdocs seek more support and specialized training

Finally, we asked respondents about their interest in possible interventions that could help increase interest in pursuing academic research careers. Postdoc respondents were asked: "Would any of the following increase your likelihood to pursue an academic research career?" Then respondents were asked to rate how strongly they disagreed or agreed with six items (*Figure 7*; *Figure 7—figure supplement 1*). In all of the statements, a higher percentage of URM postdocs expressed interest in a specialized course, training, or fellowship. While many acknowledged that interventions such as having a defined mentor outside of their lab would be helpful, URM postdocs overwhelmingly indicated the need for other support as well. These data suggest that URM postdocs are seeking more supplemental support and these interventions may serve as viable ways to encourage them to pursue research careers in academia.

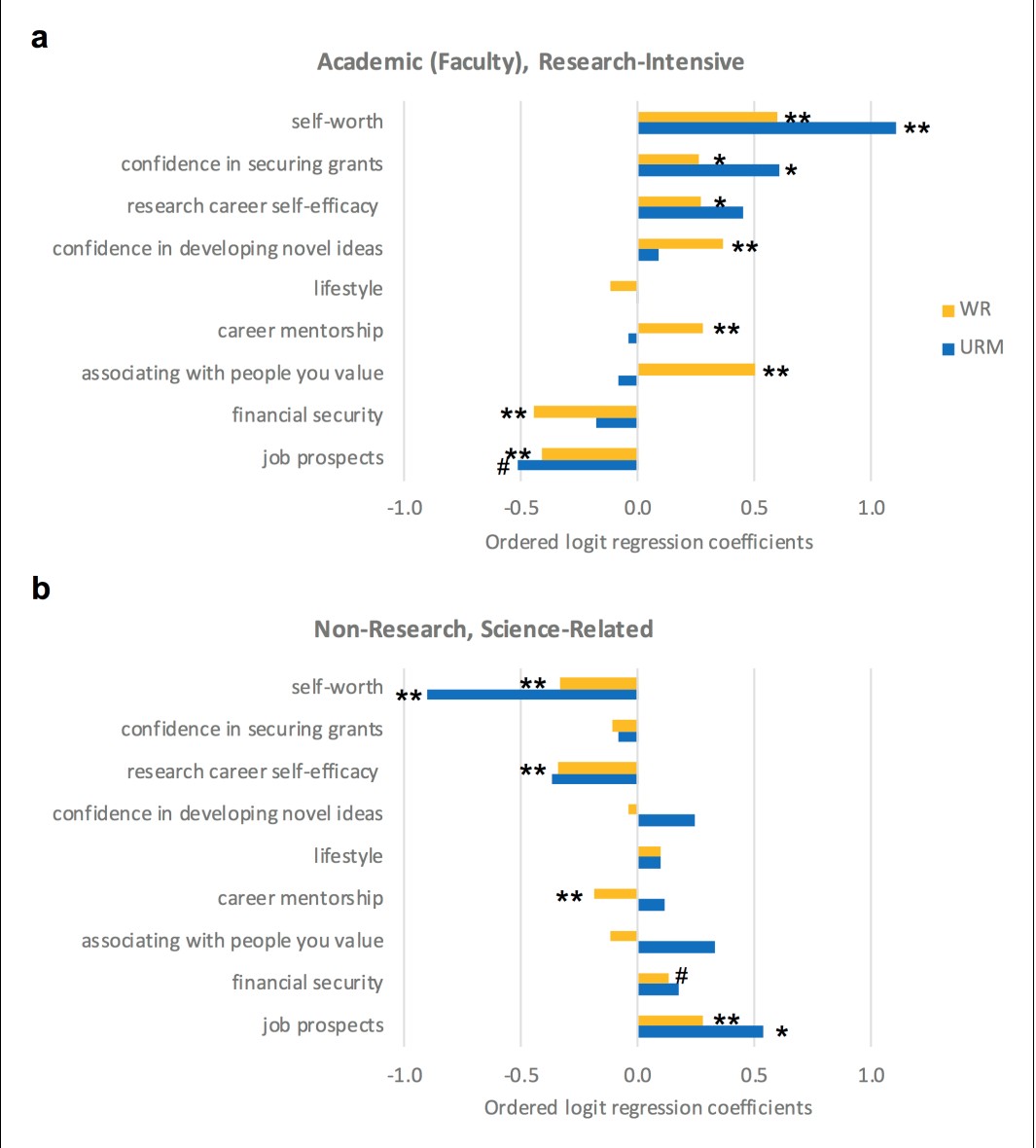

**Figure 5.** Factors that predict the career choices of underrepresented minority postdocs. Ordered logit regression analysis of factors that predict the career intentions of well-represented (yellow; WR) and underrepresented (blue; URM) postdocs intending to pursue (a) academic, research-intensive or (b) non-research, science-related careers. Associating with people they value was a strong predictor for WR postdocs choosing to pursue an academic career. Whereas, URM postdocs felt they were less likely to associate with people they value in academia. The results of the analysis are discussed in more detail in the main text. p-values represent whether the factor is significant with respect to the intention (ranging from least likely to most likely) to pursue the career path listed in the figure sub-header. **$p<0.001$, *$p<0.01$, #$p<0.05$.

## Discussion

The postdoc stage is a major point of divergence for many researchers. If institutions, and postdocs alike, are to maintain pathways for faculty diversity, the factors that influence disparate career choices must be thoroughly analyzed for potential intervention. The findings of this study help to elucidate motivations to persist at this stage of the academic path, particularly for female and URM postdocs in the biomedical and biological sciences. We reveal four main outcomes that were largely undefined in the field.

First, we report that a large percentage (nearly 60%) of biomedical and biological postdocs intend to pursue faculty positions in academia (whether in teaching or research) and an

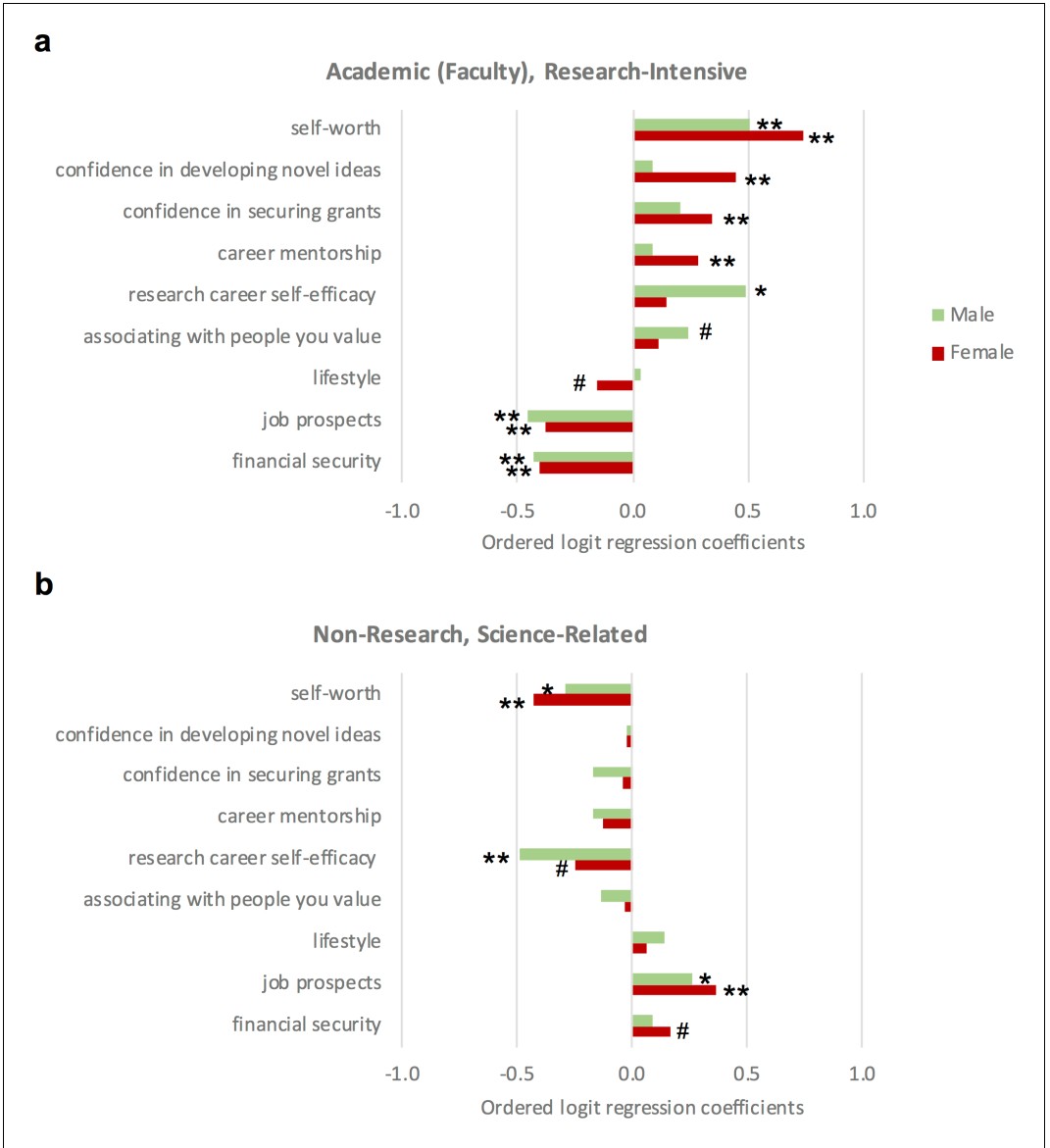

**Figure 6.** Factors that predict the career intention of male and female postdocs. Ordered logit regression analysis of factors that predict the career intentions of male (green) and female (red) postdocs intending to pursue (**a**) academic, research-intensive or (**b**) non-research, science-related careers. For female postdocs, confidence in developing novel ideas, confidence in securing grants, and career mentorship were stronger predictors for pursing a research-intensive career than for male postdocs. Whereas, female postdocs perceived lifestyle as a negative aspect for pursing a research-intensive career, it was not a predictor for why male postdocs pursue a career in academia. The results of the analysis are discussed in more detail in the main text. p-values represent whether the factor is significant with respect to the intention (ranging from least likely to most likely) to pursue the career path listed in the figure sub-header. \*\*$p<0.001$, \*$p<0.01$, #$p<0.05$.

additional 28% intend to pursue other research-intensive positions. Like McConnell et al., we find that research-intensive faculty positions still remain the most common primary career goal for postdocs (*McConnell et al., 2018*). With the limited availability of faculty positions and the changing career landscape, one might expect this number to be lower. Particularly, our

findings show that most postdocs, including URM and female postdocs, change career goals during the postdoc as opposed to starting the postdoc intending to pursue careers outside of academia. In the first year of postdoctoral training, 49% and 47% of female and URM postdocs, respectively, intend to pursue research faculty positions. This drops 10% for both groups by

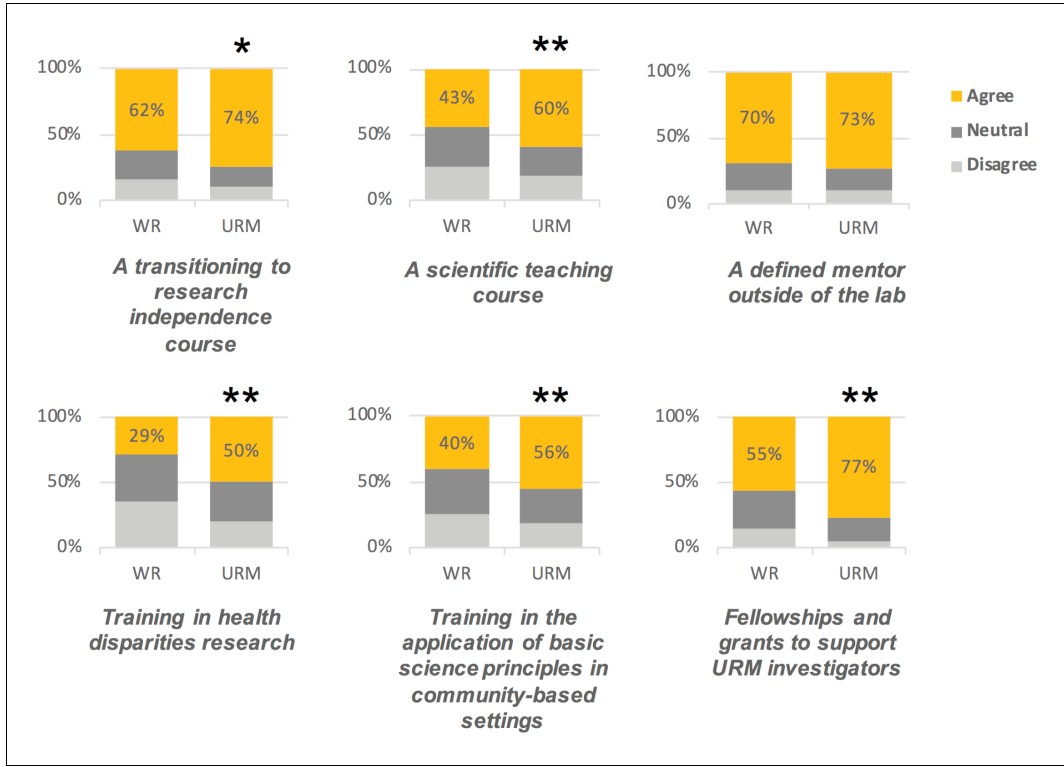

**Figure 7.** Underrepresented minorities seek additional and more specialized training. Respondents were asked whether the following factors would increase their likelihood of pursuing a research career in academia. For the purpose of the figures, 'agree' represents an aggregate of 'strongly agree' and 'agree' responses, and 'disagree' represents an aggregate of 'strongly disagree' and 'disagree' responses. URM respondents indicated that receiving more specialized courses, and support outside of the laboratory would make them more likely to pursue academic research careers.**p<0.001, *p<0.01.

The online version of this article includes the following figure supplement(s) for figure 7:

**Figure supplement 1.** Male and female perceptions of specialized training and support.

the beginning of year 3 (compared to a 3.5% increase for well-represented males). Thus, the first two years of postdoctoral training are integral for keeping female and URM postdocs in the academic research pipeline.

We also show that while perceptions around job prospects is an influential factor, the likelihood of postdocs choosing an academic research career increases as financial security, mentorship from their PI and their sense of self-worth increase. Self-worth (the sense of one's own value or worth as a person) is a rather understudied concept in understanding biomedical research training, but one's ability to achieve tasks is directly linked to one's perception of themselves and can be supported by those around them. A high sense of self-worth in the context of a research career could be a result of successful grants, publications, and the feeling that you are of high value to your field or the larger scientific workforce. Alternatively, the self-

worth theory of motivation suggests that some postdocs tend to protect their sense of self-worth by avoiding failure and risk (such as avoiding applying for competitive grants, top publications, or even faculty positions; *Covington, 1984*; *Covington and Beery, 1976*). Thus, these postdocs try to maximize their career success and avoid failure as a way to protect self-esteem. Developing structured and practical ways for postdocs to reinforce their self-worth will be important in the first few years of postdoctoral training. Mentoring and coaching models that lead to success in grants and publications should be tested for their effect on self-worth.

Upon further exploration, this study also revealed substantial disparities in self-efficacy and outcome expectations for female and URM postdocs. Women, on average reported lower expectations about the outcomes of their research efforts and lower self-efficacy around

research, which could be heavily influenced by the prevalence of gender bias in science. Female scientists can face sexism and gender bias throughout their careers that are well-documented in the literature, including pay gaps (*Nature, 2018*; *McConnell et al., 2018*), bias from science faculty (*Moss-Racusin et al., 2012*; *Sheltzer and Smith, 2014*) and gender bias in letters of recommendation (*Dutt et al., 2016*).

For URM postdocs, we suspect that the stage of training and class of the research institutions play a role in the high research self-efficacy finding (i.e. a sample of highly successful and exceptionally resilient postdocs). Importantly, we show that increasing research self-efficacy, specifically for female and URM postdocs, causes a rise in the rate of first author publications. Previous literature reminds us that individuals tend to perform at high levels when their self-efficacy is high (*Chemers et al., 2001*; *Lent et al., 2017*). Thus, we suggest postdoc and faculty development models focus on efforts that increase research self-efficacy. Successful self-efficacy workshops have been designed for biomedical graduate students and postdocs based on self-efficacy sources as outlined by Bandura: performance accomplishments, vicarious experiences (increasing role models), verbal persuasion, and physiological states (*Bakken et al., 2010*; *Bandura, 1978*).

Perhaps the most surprising aspect of this study's findings lie in the differences of career mentorship and social connectedness among URM and well-represented postdocs. The data suggests the mentoring experience for URM postdocs is perceived differently from well-represented postdocs, affecting the likelihood of pursuing academia. More than 70% of the postdocs from our study are seeking a defined mentor outside of the laboratory, suggesting that a multiple mentor, coaching model (*Williams et al., 2017*; *Williams et al., 2016*) or cascade mentorship (*Feldon et al., 2019*) could be very valuable tools to overcome differential training. Many institutions have recognized the value of mentorship training, such as the Entering Mentoring curricula, as a way to promote the success of all trainees (*Pfund et al., 2014*). We suspect that postdocs with strong, culturally-sensitive mentoring relationships will be more satisfied with their training experience and find research careers more desirable. We intend to better understand the impact of mentor-mentee relationships to career choice in future studies.

Similarly, for social connectedness, the likelihood of URM postdocs choosing non-research careers seems to increase with the belief that they will associate with people that they value most. Our data suggests that URM postdocs seeking academic research careers have not come to expect the same valued social network at the faculty level that they might have in other training periods or in other careers. Whittaker, Montgomery, and Martinez Acosta say it best: "Not being respected or actively included in [predominantly white institutions] eventually reinforces negative stereotypes and reduces self-efficacy to the point where URM postdocs can begin to question their own relevance or sense of belonging" (*Whittaker et al., 2015*). Many URM scientists know from their current academic environment that the type and level of social support that they receive is critical to their success. That support doesn't always need to come from people of similar backgrounds, but it needs to be unbiased and genuine. Institutions could be losing female and URM postdocs because they do not find people who provide this level of support on their campuses. Utilizing more cohort and structured programs for development and building community at the postdoctoral and faculty level might prove to be effective in combating these perceptions.

As is evident from *Figure 7*, underrepresented postdocs want support in more than just the laboratory so that they have the freedom to make career choices that fit their talents and values. Programs such as the Institutional Research and Career Development Awards (IRACDA), which use a structured, cohort model to support the development of postdoctoral scholars have been successful (*Eisen and Eaton, 2017*; *Rybarczyk et al., 2016*), but exist at only a select number of institutions. Even still, if postdocs find community among themselves but not the faculty at institutions in which they seek to join, these programs do not solve the problem. Our study suggests a strong role for interventions that begin to both increase institutional diversity and move toward a more inclusive culture, particularly for postdoctoral fellows at a critical junction in their careers.

## Methods

### U-MARC Survey

Postdoctoral scholars (postdocs) in the biological and biomedical sciences were asked to complete an original survey entitled U-MARC (Understanding Motivations for Academic Research Careers; *Supplementary file 1*). The

70-item survey (1) assesses participants views on factors associated with careers in science, and (2) measures outcome expectations and self-efficacy around research careers using two original scales. Our study's theoretical framework is based in (i) Social Cognitive Career Theory which states that self-efficacy, outcome expectations, and personal goals influence career decision and (ii) Vroom's Expectancy Theory which assumes that motivation is an outcome of how much an individual wants a reward (valence), the likelihood that a particular effort will lead to the expected performance (expectancy) and the belief that the performance will lead to the reward (instrumentality) (*Lent et al., 1994*; *Vroom, 1964*). If the outcomes available from high effort and high performance are not attractive to the individual, motivation to perform that behavior will be low. We used expectancy theory to develop the outcome expectations instrument in the U-MARC survey, with some items taken from the Research Outcome Expectations Questions (ROEQ) (*Bieschke, 2000*). We used a specific scale for self-efficacy that determined respondents' belief in their ability to achieve research-orientated tasks such as grants, publications, mentorship, and a research career (distinct from outcome expectations). All items were subjected to content analysis using expert judges. Pre-testing was conducted with members of the target population and items were refined and modified.

### Data collection and procedures

All work was done under the approval of the Weill Cornell Medicine Institutional Review Board (IRB# 1612017849), and all respondents provided consent for participation in the study. A purposeful sampling strategy was used where participants were recruited through postdoctoral listservs from many top-ranked research universities and institutions (*Supplementary file 2*). Snowball sampling was discouraged. In most instances, the recruitment email was forwarded twice by a postdoctoral listserv manager to their respective postdocs. In some cases, listservs included other scientists in addition to postdocs, but those not fitting the inclusion criteria were excluded. Only respondents who identified as a postdoctoral scholar or postdoctoral research associate were included in the analysis (n = 1284). However, out of these 1284 respondents, 36 did not specify a race, ethnicity or gender and could therefore not be included in the analysis of URM, WR, male and female groups. MD and MD-PhD recipients were excluded from

the analysis. Sampled postdoc participants represent 6% of the total pool (21,781) of appointed biomedical and biological postdocs the year the survey was conducted (2017) according to the National Science Foundation.

Representation was sought from a variety of biomedical fields across the US (*Supplementary file 2*), with an overrepresentation of postdocs who were US citizens or permanent residents (n = 679, 53%), female postdocs (n = 774, 60%) and URM postdocs (n = 174, 14%). However, 7 female and 2 URM postdocs did not indicate a career choice and were therefore excluded from the analysis shown in *Figure 1d* and the accompanying source data. Underrepresented minorities include the racial categories of American Indian or Alaska Native, Black or African American, Native Hawaiian or Other Pacific Islander and/or the ethnicity of Hispanic or Latino. Well-represented respondents identified as Asian or White and Non-Hispanic or Latino. The sample also represents wide geographic (over 80 universities) and subfield diversity. The 70-item anonymous U-MARC survey instrument was collected and managed using REDCap electronic data capture tools. REDCap (Research Electronic Data Capture) is a secure, web-based application designed to support data capture for research studies.

In the survey, respondents were asked to rate their interest in pursuing each of the following career pathways: (i) Academic (Faculty), Research Intensive, (ii) Academic (Faculty), Teaching Intensive, (iii) Other Research Intensive, (iv) Non-Research, Science-Related, or (v) Non-Science Related as well as complete self-efficacy and outcome expectations instruments using a five-point Likert scale. Non-Research, Science-Related careers are defined as positions that are not actively involved in research but are still science-related, such as science/technical writing, science administration and management, science communication, technology transfer, and scientific teaching (K-12). Non-Science Related careers are positions not conventionally related to science, such as real estate, finance or public policy careers. They were then asked to agree or disagree with statements about influential factors using the five-point Likert scale, followed by statements regarding outcome expectations and research self-efficacy (*Supplementary file 1*).

### Statistical analysis

In the Results section, three types of regression techniques are applied. For models of the number of first-author publications in *Table 3*,

negative binomial regressions (*Long, 1997*) that account for over dispersion are applied and the estimates are incidence-rate ratios. The regression for publication rate in *Table 3* uses an ordinary least squares (OLS) multiple regression model with a logarithmic transformation to account for skewness. Since the regression models for the publication rate is in a logarithmic scale for the dependent variable, mean changes can be calculated by exponentiating the underlying regression coefficients. The exponentiated regression coefficients are displayed as mean changes in *Table 3*. Residual analysis suggests that the logarithmic transformation of first-author publication rate yields a model that satisfies the OLS modeling assumptions.

A number of analyses regarding perceptions/opinions use ordinal logistic regression (*Agresti, 2002*) since the postdoc respondents were asked to rate influential factors to their career choice on a five-point Likert-type scale (strongly disagree, disagree, neutral, agree, and strongly agree). In some of the regression models, the respondents were stratified by gender and URM status. The regression models were fit using *Stata 15* (*StataCorp, 2017*). Chi-Square ($\chi^2$) analyses were used to test for significantly different proportions of demographic groups within each career interest track.

### Acknowledgements
We thank Mary Charlson and Carla Boutin-Foster for critical feedback on the study design, and Ushma Neil and Louise Hainline for helpful comments on the manuscript.

**W Marcus Lambert** is in the Department of Medicine and Weill Cornell Graduate School of Medical Sciences, Weill Cornell Medicine, New York, United States

wil2009@med.cornell.edu

https://orcid.org/0000-0002-4592-5176

**Martin T Wells** is in the Department of Statistics and Data Science, Cornell University, Ithaca, United States

**Matthew F Cipriano** is in the Weill Cornell Graduate School of Medical Sciences, Weill Cornell Medicine, New York, United States

**Jacob N Sneva** is in the Weill Cornell Graduate School of Medical Sciences, Weill Cornell Medicine, New York, United States

**Juanita A Morris** is in the Teachers College, Columbia University, New York, United States

https://orcid.org/0000-0001-9246-9866

**Linnie M Golightly** is in the Department of Medicine, Weill Cornell Medicine, New York, United States

**Author contributions:** W Marcus Lambert, Conceptualization, Resources, Data curation, Software, Formal analysis, Supervision, Funding acquisition, Validation, Investigation, Visualization, Methodology, Project administration; Martin T Wells, Data curation, Software, Formal analysis, Validation, Methodology; Matthew F Cipriano, Jacob N Sneva, Conceptualization, Formal analysis, Investigation, Methodology; Juanita A Morris, Conceptualization, Formal analysis, Investigation; Linnie M Golightly, Conceptualization, Supervision, Funding acquisition, Methodology, Project administration

**Competing interests:** The authors declare that no competing interests exist.

**Ethics:** Human subjects: All work was done under the approval of the Weill Cornell Medicine Institutional Review Board (IRB# 1612017849), and all respondents provided consent for participation in the study.

### Funding

| Funder | Grant reference number | Author |
| --- | --- | --- |
| National Center for Advancing Translational Sciences | Award Number UL1TR002384 | W Marcus Lambert Matthew F Cipriano Jacob N Sneva Linnie M Golightly |
| National Institute of Allergy and Infectious Diseases | K24 AI110732 | Linnie M Golightly |

The funders had no role in study design, data collection and interpretation, or the decision to submit the work for publication.

**Decision letter and Author response**
Decision letter https://doi.org/10.7554/eLife.48774.sa1
Author response https://doi.org/10.7554/eLife.48774.sa2

## Additional files

### Supplementary files
- Supplementary file 1. Survey questionnaire.
- Supplementary file 2. Top US life science research institutions and number of respondents by institution.
- Transparent reporting form

### Data availability
To protect the privacy and confidentiality of the research participants, the raw datasets are not available in online databases. Summaries of the data

analyzed in this study are included in the manuscript and supporting files.

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
