## [Decision Letter]

Summary:

This paper seeks to understand how external and internal factors play a role in career choices, and hence diversity of the biomedical scientific landscape. While the paper has lots of potential and certainly can make an impact on how institutions train, support and engage postdoctoral fellows, there major and minor concerns that we hope can be addressed.

Major comments to address:

1) With regard to framing, there were missed opportunities to discuss imposter syndrome, gender bias, intersectionality, and diversity/cultural expectations with greater clarity and specificity. Most of what is written in the introduction and conclusion is framed around career outcomes and trends in academia; however, I think it is important to spend the same energy introducing what other scholars have found with regard to the issues women and under-represented minorities face in the biomedical sciences. For example, how the female responses to publication rate, mentorship and financial security link to issues such as publication bias, finding mentors as a female/minority, and the gender pay gap.

Authors should also expand on why institutes lose URMs and women as a result of them not finding people they value on their campus; how does the lack of these social networks in their current institute discourage them from applying for academic positions elsewhere? Additionally, the results on possible interventions that could help increase interest in pursuing academic research careers are only presented for WR and URM. It would be interesting to also include the scores of male and female respondents (both WR and URM) to see how they compare.

2) At the start of the articles the authors clearly describe how the number of biomedical doctoral trainees and postdocs far exceeds the number of faculty position, and that the odds of obtaining a position are low. However, there are missed opportunities to discuss how because of this 1) students/postdocs need to be better prepared for non-academic careers, such as through internships/work experience; 2) Why it's important to dispel the myth that non-traditional careers are second class; 3) Explain how the data from the survey suggests that career development offices are no longer added value, but essential.

3) Re the survey and statistical analyses i) Please include the full survey questions in supplemental materials. (essential)

ii) Any time the authors state that there is a difference between two groups of people they need to provide the statistical analysis (N, test statistic, and p-value) that shows there is a statistically significant difference between these two groups. (essential)

iii) The number of respondents in each group needs to be supplied either on the figures or in a table. For example, how many of the respondents are URM/WR/Female/Male and how many respondents are in each of the career intention categories. (essential)

iv) In Results subsection “Predictors of career sector among biomedical postdocs”: Ordered regression analyses are often influenced by the order which factors are entered. I suggest the authors also try a non-ordered regression analysis, or multiple permutations of the ordered regression analysis, to ensure order is not skewing their results. (optional)

v) In the Methods subsection “U-MARC Survey”: The authors do not describe to whom (directly to target group emails, blanket emails to listservs that includes everyone in the institution, or via an institutional office, etc.), and the frequency by which (how many times were respondents asked to complete the survey) the instrument was distributed. They also do not provide the numbers by institution, minority status, and gender so that the reader understands how the distribution of these number may impact the results presented. (optional)

vi) Figure 3: Due to multiple testing, a p-value of 0.05 is not significant. Please adjust using a Bonferonni correction (essential)

vii) Table 2: The correlation between research self-efficacy and publications is very weak (note regression coefficients). In fact, one would usually call that a small, yet significant effect. The authors should adjust their results section accordingly (essential)

viii) Statistical analyses: I suggest the authors add gender*URM to the regression models, to see if there is an interaction effect. (optional)

ix) Given how integrated these questions are, and the likelihood that many of the responses are correlated, I suggest the authors conduct a model (either logit or nominal logistic) to determine which of these factors are actually significantly different between those interested in academia and those interested in other careers. If these factors were included in the model the authors ran, then that needs to be stated explicitly (please list all the factors included in the model). (optional)

4) You mention intersectionality in the introduction, but the presentation of your data doesn't consider intersectionality at all. It is strictly based on non-homogenous groupings of postdoctoral fellows by minority status and gender.

5) Mentions of 'alternative careers', 'leave science' and 'other' careers in the introduction and results are written in a manner that perpetuates the idea that non-research intensive academic careers are somehow second rate, and should be re-phrased. Please use: non-academic careers, careers outside academia, or other career tracks.

6) Issues with Figure 1C and D:

i) The number of post-doc respondents for each year should be supplied in Figure 1D

ii) Data should be broken down into: WR-male, WR-female URM-male, URM-female. Does the data for women encompass both URM and WR?

iii) For Figure 1C could the increase in intention to stay in academia years 3-7 be because most postdocs not interested in academia have already left - is that data surprising?

iv) In the Results subsection “Academia remains a top choice for biomedical postdocs": Is the "term limit" the only factor for this drastic decline? Seems like an artefact of the data.

7) To what extent did authors control for post-doc experience (number of year working as a postdoc) in analysing and reporting on the factors influencing career intentions? Studies in Europe (Germany, Chlosta et al 2010, and The Netherlands, van der Weijden et al 2016) revealed that after a longer duration of their temporary contracts, postdocs in both countries are less likely to invest in a risky academic career, and this could be influencing the results shown. For example, in the Results subsection “Academia remains a top choice for biomedical postdocs" it is unclear if the percentages of committed vs uncommitted are valid as the authors have not provided the distribution (year) and numbers in each year of respondents. Are the numbers equally distributed (by year, URM status, gender)?

8) Interpretation of responses to, 'How influential was your PI on your postdoctoral career', are misleading, as it is unclear how the author links 'influence' to 'lack of guidance (Results subsection “Postdocs lack career guidance from mentors”). Unless there are follow up questions not included in the article, interpretation of the results should be re-phrased to more accurately reflect the question respondents were asked.

9) Figure 3b: Wow! Is this result consistent with other studies?

10) In the Results subsection “Predictors of career sector among biomedical postdocs”: What do the authors mean by "..not value those in the academy to the same degree as men."? Do they mean that women had a lower score for the statement "Pursuing a research-intensive faculty position would enable me to associate with the kind of people that I value most" than men? Please be specific- there may be other co-factors at work here, such as the fact that men are more represented than women in academia.

11) What are some ways/interventions the authors might suggest to support increasing URM's self-worth as part of their training in the first 2 years? Newer than Bandura 1978?

12) In the Discussion: Here and throughout, it would be helpful if the authors were specific about where they grouped women in the context of under represented in science. Sometimes it sounds like they are grouped with well-represented, and others as under represented. An increase in specificity will make this easier to understand.

13) When assessing 'self-efficiency' many of the questions asked by the authors were more about interest in a research-intensive career than whether one can be successful at it. We often associate self-efficacy with self-confidence in one's ability to exert control over one's own motivation, behavior, and accomplishments, and if these two groups of postdocs (those interested in academia and those not interested in academia) do not have the same goals, we cannot use goal-specific questions to assess expectancy and instrumentality. This should be discussed in the paper, or the authors should consider using a different term from self-efficiency.

14) In the Results subsection “Outcome expectations and self-efficacy among women and URM postdocs”: The authors spend lots of time teasing out the differential perceptions of factors influential to choosing a career. However, the analyses for women and URMs seems an afterthought, especially in light of how the introduction and discussion are currently framed. Why are the data for URMs and women not presented the same way as in Figure 2?

15) Many of results discussed in the second paragraph of the discussion are almost identical to those that were reported in McConnell et al., 2018. The similarities between these two studies should be discussed more explicitly.

---

## [Author Response]

[We repeat the reviewers’ points here in italic, and include our replies point by point, as well as a description of the changes made, in Roman.]

Major comments to address:1) With regard to framing, there were missed opportunities to discuss imposter syndrome, gender bias, intersectionality, and diversity/cultural expectations with greater clarity and specificity. Most of what is written in the introduction and conclusion is framed around career outcomes and trends in academia; however, I think it is important to spend the same energy introducing what other scholars have found with regard to the issues women and under-represented minorities face in the biomedical sciences. For example, how the female responses to publication rate, mentorship and financial security link to issues such as publication bias, finding mentors as a female/minority, and the gender pay gap.Authors should also expand on why institutes lose URMs and women as a result of them not finding people they value on their campus; how does the lack of these social networks in their current institute discourage them from applying for academic positions elsewhere? Additionally, the results on possible interventions that could help increase interest in pursuing academic research careers are only presented for WR and URM. It would be interesting to also include the scores of male and female respondents (both WR and URM) to see how they compare.

Reply: We have taken a different approach to framing the introduction. Instead of starting with so much focus on career outcomes and trends in academia, we spend more time addressing the disparities in career choice that exist for female and underrepresented minority scientists (e.g. a decreasing interest in academia). In the discussion, we expand on possible explanations for why academic institutions lose URM and women scientists (differential mentorship and non-inclusive environments). We did not find significant differences for Figure 7 among male and female respondents. This is now included in Figure 7 – figure supplement 1.

2) At the start of the articles the authors clearly describe how the number of biomedical doctoral trainees and postdocs far exceeds the number of faculty position, and that the odds of obtaining a position are low. However, there are missed opportunities to discuss how because of this 1) students/postdocs need to be better prepared for non-academic careers, such as through internships/work experience; 2) Why it's important to dispel the myth that non-traditional careers are second class; 3) Explain how the data from the survey suggests that career development offices are no longer added value, but essential.

Reply: We have removed the focus on postdoctoral trends and the odds of obtaining a faculty position, and instead focus the introduction more on examining the declining interest in academic careers by postdoctoral trainees from underrepresented groups. We do however reiterate the reviewer’s suggested messages in the introduction.

3) Re the survey and statistical analysesi) Please include the full survey questions in supplemental materials. (essential)

Reply: We have now included the full survey.

ii) Any time the authors state that there is a difference between two groups of people they need to provide the statistical analysis (N, test statistic, and p-value) that shows there is a statistically significant difference between these two groups. (essential)

Reply: We have addressed all statistical differences in the Methods section, Figures, and Supplementary files.

iii) The number of respondents in each group needs to be supplied either on the figures or in a table. For example, how many of the respondents are URM/WR/Female/Male and how many respondents are in each of the career intention categories. (essential)

Reply: We have included the number of URM and female respondents in the text and provided additional supplementary tables and figures illustrating (1) the breakdown of respondents in each career intention category, (2) the total number of respondents by year and by gender and underrepresentation, and (3) the number of respondents most likely to pursue academia by year, gender and underrepresentation.

iv) In Results subsection “Predictors of career sector among biomedical postdocs”: Ordered regression analyses are often influenced by the order which factors are entered. I suggest the authors also try a non-ordered regression analysis, or multiple permutations of the ordered regression analysis, to ensure order is not skewing their results. (optional)

Reply: The ordinal regression was applied in the setting where the dependent variable levels were determined by the Likert scale’s values. There was no influence on the results by different ordering of the independent variables.

v) In the Methods subsection “U-MARC Survey”: The authors do not describe to whom (directly to target group emails, blanket emails to listservs that includes everyone in the institution, or via an institutional office, etc.), and the frequency by which (how many times were respondents asked to complete the survey) the instrument was distributed. They also do not provide the numbers by institution, minority status, and gender so that the reader understands how the distribution of these number may impact the results presented. (optional)

Reply: We have updated the methods section with more detail on the method and frequency of distribution to each institution. (Methods subsection “Data Collection and Procedures”). We have included a Supplementary File 2 listing the institutions of the respondents, the percentage of respondents from each institution, and we have included in the methods the numbers of respondents by gender and minority status.

vi) Figure 3: Due to multiple testing, a p-value of 0.05 is not significant. Please adjust using a Bonferonni correction (essential)

Reply: In the Results subsection “Outcome expectations and self-efficacy among female and URM postdocs” we added a sentence that clarifies our cautious interpretation of the result in the presence of multiple comparisons. Strict Bonferroni correction is not significant, but it is surprising to us that the research self-efficacy of URM postdocs trends in this direction, especially given the lower first author publication rate.

vii) Table 2: The correlation between research self-efficacy and publications is very weak (note regression coefficients). In fact, one would usually call that a small, yet significant effect. The authors should adjust their results section accordingly (essential)

Reply: In Table 2, the original coefficients that we provided made the values seem very small, as these were the output from a negative binomial regression and linear regression with the log rate as the dependent variable. We have now transformed these coefficients for first author publications to incidence rate ratios and the exponentiated coefficients and updated the manuscript text and Table 2 accordingly.

viii) Statistical analyses: I suggest the authors add gender*URM to the regression models, to see if there is an interaction effect. (optional)

Reply: There is no significant interaction effect from gender and URM (p=0.371) in Table 2 or in our other regression models. It is likely that there are too few URM postdocs to see a significant effect, a phenomenon observed by other researchers recruiting URM postdocs in the biomedical sciences. Gibbs, McGready, and Griffin 2015 note: “While the sample of postdocs from URM backgrounds is large, the sample of URMM is sometimes underpowered and not able to adequately capture potential differences between their responses and those of postdocs from other groups.”

ix) Given how integrated these questions are, and the likelihood that many of the responses are correlated, I suggest the authors conduct a model (either logit or nominal logistic) to determine which of these factors are actually significantly different between those interested in academia and those interested in other careers. If these factors were included in the model the authors ran, then that needs to be stated explicitly (please list all the factors included in the model). (optional)

Reply: The regression models that are presented in the manuscript (i.e. Table 2) list all of the variables that are included in the models.

4) You mention intersectionality in the introduction, but the presentation of your data doesn't consider intersectionality at all. It is strictly based on non-homogenous groupings of postdoctoral fellows by minority status and gender.

Reply: Like other studies assessing biomedical/biological postdocs (Gibbs, McGready, and Griffin, 2015), the sample of URM males remains underpowered. We therefore present most of the data aggregated by minority status and gender. We have removed the intersectionality phrase.

5) Mentions of 'alternative careers', 'leave science' and 'other' careers in the introduction and results are written in a manner that perpetuates the idea that non-research-intensive academic careers are somehow second rate, and should be re-phrased. Please use: non-academic careers, careers outside academia, or other career tracks.

Reply: Thank you for pointing this out. We have re-phrased the language where possible other places that refer to non-academic careers as ‘other’.

6) Issues with Figure 1C and Di) The number of post-doc respondents for each year should be supplied in Figure 1D.

Reply:We have included Figure 1 – figure supplement 1 indicating the number of postdoc respondents most likely to pursue academia, by year, gender and underrepresentation.

ii) Data should be broken down into: WR-male, WR-female URM-male, URM-female. Does the data for women encompass both URM and WR?

Reply: The sample of URM postdocs is too small to disaggregate the data in this way. ‘Women’ include both well-represented and underrepresented postdocs and ‘URM’ include both postdocs identifying as male and female.

iii) For Figure 1C could the increase in intention to stay in academia years 3-7 be because most postdocs not interested in academia have already left - is that data surprising?

Reply: Previous literature seems to support that hypothesis. Silva et al, 2016 showed that more UCSF postdocs who attained non-faculty positions did so after 2 years of training. The data for postdocs who were now in faculty positions skewed toward longer training times. Kahn and Ginther 2017, also showed that postdocs pursuing industry and other non-academic positions spend less time in their postdoctoral training. We thank you for pointing this out and have included this information in the text.

iv) In the Results subsection “Academia remains a top choice for biomedical postdocs": Is the "term limit" the only factor for this drastic decline? Seems like an artefact of the data.

Reply: In Figure 1d, we observed this trend for both WR males and URMs (which include male and female postdocs). The sample size for WR males in year 6 (n=26) is smaller than the number of postdocs in years 1-5, but this is consistent with the steady decline of respondents by length of training. The sample is also consistent with other years in terms of representation (institution, sub-field, marital status). We felt that it was important to include data from later years as this could be reflective of what is occurring as a result of postdoctoral term limits. We provide the number of respondents per year in Figure 1 – figure supplement 1 and note this in the caption of Figure 1.

7) To what extent did authors control for post-doc experience (number of year working as a postdoc) in analysing and reporting on the factors influencing career intentions? Studies in Europe (Germany, Chlosta et al 2010, and The Netherlands, van der Weijden et al 2016) revealed that after a longer duration of their temporary contracts, postdocs in both countries are less likely to invest in a risky academic career, and this could be influencing the results shown. For example, in the Results subsection “Academia remains a top choice for biomedical postdocs" it is unclear if the percentages of committed vs uncommitted are valid as the authors have not provided the distribution (year) and numbers in each year of respondents. Are the numbers equally distributed (by year, URM status, gender)?

Reply: We have included Figure 1-figure supplement 1-source data 1 showing the numbers and distribution by year. 54% of *early postdocs* (<3 years) were less committed to academia, compared to 65% of *late postdocs* (>3 years). We have revised Figure 1e to now include pie graphs for early and late postdocs. There was only a 2% difference between the WR and URM postdoc percentages who were less committed, and a 4% difference between male and female postdocs who were less committed to academia. The differences between early and late postdocs regarding ‘the factors that influence career intentions’ were not large enough to alter the trends shown throughout the manuscript.

8) Interpretation of responses to, 'How influential was your PI on your postdoctoral career', are misleading, as it is unclear how the author links 'influence' to 'lack of guidance (Results subsection “Postdocs lack career guidance from mentors”). Unless there are follow up questions not included in the article, interpretation of the results should be re-phrased to more accurately reflect the question respondents were asked.

Reply: The statement given to the survey respondents specifically asks about guidance: “Guidance from my lab PI has highly influenced which career path I will pursue.” We have updated the phrasing in the text to reflect this. The authors apologize for this confusion.

9) Figure 3b: Wow! Is this result consistent with other studies?

Reply: Gibbs, McGready, & Griffin, 2015 found a similar result, with 87% of URM males vs. 83% WR males agreeing with the statement, “I am confident in my abilities as an independent researcher.” However, unlike our findings, they found that URM women had lower research self-efficacy than WR women.

10) In the Results subsection “Predictors of career sector among biomedical postdocs”: What do the authors mean by "..not value those in the academy to the same degree as men."? Do they mean that women had a lower score for the statement "Pursuing a research-intensive faculty position would enable me to associate with the kind of people that I value most" than men? Please be specific- there may be other co-factors at work here, such as the fact that men are more represented than women in academia.

Reply: Yes, women had a lower score compared to men for the statement, “Pursuing a research-intensive faculty position would enable me to associate with the kind of people that I value most.” We clarified this in the text (Results subsection “Factors that predict the career choice of postdocs”).

11) What are some ways/interventions the authors might suggest to support increasing URM's self-worth as part of their training in the first 2 years? Newer than Bandura 1978?

Reply: There is a wealth of literature examining postgraduate students’ self-efficacy (which is linked to Bandura), but few studies show effective interventions. In most reports, a student’s experience of succeeding in research tasks is seen as the most important source of self-efficacy beliefs. We’ve included suggestions based on these empirical studies in our discussion.

Self-worth (the sense of one's own value or worth as a person) is a rather understudied concept in understanding biomedical graduate training. It may be that postdocs with high self-worth value themselves and their worth to the scientific workforce or academy because their colleagues also see them as valuable and demonstrate it through verbal affirmations, inclusion, and sponsorship (external influences). We expand on this in the discussion.

12) In the Discussion: Here and throughout, it would be helpful if the authors were specific about where they grouped women in the context of under represented in science. Sometimes it sounds like they are grouped with well-represented, and others as under represented. An increase in specificity will make this easier to understand.

Reply: Consistent with NIH definitions, we define “underrepresented minority” (URM) as individuals who identify as Blacks or African Americans, Hispanics or Latinos, American Indians or Alaska Natives, and/or Native Hawaiians or other Pacific Islanders. We use “underrepresented groups” only three times throughout the manuscript to include both underrepresented minorities and women, since women have been shown to face particular challenges at the graduate level and beyond and are underrepresented at senior faculty levels in most biomedical-relevant disciplines. We specify whenever we are referring to underrepresented minorities or women, and recognize that women could include well-represented and URM postdocs.

13) When assessing 'self-efficiency' many of the questions asked by the authors were more about interest in a research-intensive career than whether one can be successful at it. We often associate self-efficacy with self-confidence in one's ability to exert control over one's own motivation, behavior, and accomplishments, and if these two groups of postdocs (those interested in academia and those not interested in academia) do not have the same goals, we cannot use goal-specific questions to assess expectancy and instrumentality. This should be discussed in the paper, or the authors should consider using a different term from self-efficiency.

Reply: We are not clear on the meaning of ‘self-efficiency’, which we see as different from ‘self-efficacy’. The reviewers are correct in that self-efficacy beliefs are goal-oriented and context-specific, but also include future-oriented judgments of capabilities that change according to the task involved. We used a specific scale for self-efficacy that determined their belief in their ability to achieve research-orientated tasks such as grants, publications, mentorship, and a research career. The latter is not about career intention or career goals, rather a statement about their belief in their ability to achieve success as a researcher (distinct from outcome expectations). We discuss this in the methods and include which questions were used in both the outcome expectations and self-efficacy scales in Figure 3 – figure supplement 1.

We think that it’s important to note that self-efficacy beliefs, outcome expectations, and goals are the primary variables in Social Cognitive Career Theory (SCCT), the theory by which our work is grounded in. SCCT posits that goals are importantly tied to both outcome expectations and self-efficacy.

14) In the Results subsection “Outcome expectations and self-efficacy among women and URM postdocs”: The authors spend lots of time teasing out the differential perceptions of factors influential to choosing a career. However, the analyses for women and URMs seems an afterthought, especially in light of how the introduction and discussion are currently framed. Why are the data for URMs and women not presented the same way as in Figure 2?

Reply: We see how the analysis for women and URM could be perceived as an afterthought. However, there are few significant differences between female and male and URM and WR postdocs regarding the factors from Figure 2. The differences were summarized in Figure 3a and 3b, and we have included the supporting data. We’ve also rewritten the introduction to focus more on the challenges faced by these underrepresented groups.

15) Many of results discussed in the second paragraph of the discussion are almost identical to those that were reported in McConnell et al., 2018. The similarities between these two studies should be discussed more explicitly.

Reply: The findings that overlap with McConnell et al., 2018 are that (1) research-intensive faculty positions still remain the most common primary career goal for postdocs, (2) a recommendation for formal trainings to increase mentorship satisfaction, and (3) gender-specific differences in career choice. We address these similarities in the discussion.